DATA RELEASE

# Mosquito alert: leveraging citizen science to create a GBIF mosquito occurrence dataset

Živko Južnič-Zonta[1], Isis Sanpera-Calbet[4], Roger Eritja[2], John R.B. Palmer[4], Agustí Escobar[2], Joan Garriga[1], Aitana Oltra[4], Alex Richter-Boix[2], Francis Schaffner[3], Alessandra della Torre[5], Miguel Ángel Miranda[6], Marion Koopmans[7], Luisa Barzon[9], Frederic Bartumeus Ferre[1,2,8,*], Mosquito Alert Digital Entomology Network[†] and Mosquito Alert Community[‡]

1 Centre d'Estudis Avançats de Blanes (CEAB-CSIC), C/d'accés a la Cala St. Francesc 14, 17300 Blanes, Girona, Spain
2 Centre de Recerca Ecològica i Aplicacions Forestals (CREAF), Edifici C Campus de, 08193 Bellaterra, Barcelona, Spain
3 Francis Schaffner Consultancy (FSC), Lörracherstrasse 50, 4125 Riehen, Switzerland
4 Departament de Ciències Polítiques i Socials, Universitat Pompeu Fabra, Plaça de la Mercè, 10-12, 08002 Barcelona, Spain
5 Department Public Health and Infectious Diseases (UNIROMA1), Sapienza University, 00185 Rome, Italy
6 University Balearic Islands, Applied Zoology and Animal Conservation Research Group (UIB), Ctra. Valldemossa km 7.5, 07122, Palma, Spain
7 Erasmus University Medical Center (Erasmus MC), Doctor Molewaterplein 40, 3015 GD Rotterdam, Netherlands
8 Institució Catalana de Recerca i Estudis Avançats (ICREA), 23 Passeig de Lluís Companys, 08010 Barcelona, Spain
9 Department of Molecular Medicine (UNIPV), Università degli Studi di Padova, 63 Via Gabelli, 35121 Padova, Italy

**Submitted:** 21 March 2022

\* Corresponding author. E-mail: fbartu@ceab.csic.es

† Collaborative Authors: Entomological experts who validated the dataset and their affiliations appears at the end of the publication

‡ Citizen scientists and community builders who actively participate in the Mosquito Alert project (www.mosquitoalert.com)

Preprint submitted at https://doi.org/10.5281/zenodo.6379002

Included in the series: **Vectors of human disease** (https://doi.org/10.46471/GIGABYTE_SERIES_0002)

## ABSTRACT

The Mosquito Alert dataset includes occurrence records of adult mosquitoes collected worldwide in 2014–2020 through Mosquito Alert, a citizen science system for investigating and managing disease-carrying mosquitoes. Records are linked to citizen science-submitted photographs and validated by entomologists to determine the presence of five targeted European mosquito vectors: *Aedes albopictus*, *Ae. aegypti*, *Ae. japonicus*, *Ae. koreicus*, and *Culex pipiens*. Most records are from Spain, reflecting Spanish national and regional funding, but since autumn 2020 substantial records from other European countries are included, thanks to volunteer entomologists coordinated by the AIM-COST Action, and to technological developments to increase scalability. Among other applications, the Mosquito Alert dataset will help develop citizen science-based early warning systems for mosquito-borne disease risk. It can also be reused for modelling vector exposure risk, or to train machine-learning detection and classification routines on the linked images, to assist with data validation and establishing automated alert systems.

**Subjects** Ecology, Taxonomy, Biodiversity

## DATA DESCRIPTION

### Background

Vector-borne diseases (VBDs) are infections caused by pathogens transmitted by carrier species (vectors), most of which are arthropods. VBDs are a major global health issue, with 80% of the world's population at risk of one or more of these diseases [1]. VBDs account for 17% of the global burden of communicable diseases, with over 1 billion infections and over 700,000 deaths caused by VBDs annually [1]. Many of these diseases, once limited to tropical and subtropical zones, are now increasingly seen in temperate areas [1, 2].

Among VBDs, mosquito-borne diseases (MBDs) account for a large share of cases. In 2017 the World Health Organization estimated over 347 million MBD cases and over 447,000 deaths caused by MBDs annually [1]. Of the 3591 known species of mosquitoes (order Diptera; family Culicidae) [3], only a fraction are involved in disease transmission or cause considerable nuisance to human and animal populations. These include invasive species that are spreading throughout Europe owing to globalization and climate change [2, 4].

There are five mosquito vectors of primary concern in Europe: four *Aedes* invasive mosquitoes (AIMs) and the native *Culex pipiens* (northern house mosquito; NCBI:txid7175). The four AIMs established in Europe are *Ae. (Stegomyia) aegypti* (yellow fever mosquito; NCBI:txid7159), *Ae. (Stegomyia) albopictus* (Asian tiger mosquito; NCBI:txid7160), *Ae. (Hulecoetomyia) japonicus* (Asian bush mosquito; NCBI:txid140438) and *Ae. (Hulecoetomyia) koreicus* (Korean bush mosquito; NCBI:txid586676) [5]. Their ability to spread into new territories, and their capacity to act as vectors of tropical viral diseases such as dengue, chikungunya, Zika, yellow fever and Japanese encephalitis, make AIMs key vectors of public health relevance [6]. Notably, *Ae. albopictus* has already caused outbreaks of exotic arboviruses in Europe, i.e. outbreaks of dengue in Croatia, France, Spain and Italy [7, 8, 9, 10], and two of chikungunya in Italy [11]. In Europe, *Culex pipiens* is considered the principal vector of West Nile virus (WNV) [12, 13] and Usutu virus [14]. Since 2010, the WNV epidemiological pattern in Europe has evolved, with an increasing incidence of human and equine cases after what began with a very low level of endemicity. Several WNV outbreaks have occurred in recent decades, and there was a significant peak in incidence in 2018, with 1503 cases in the European Union [13, 15, 16].

Given the absence of effective vaccine solutions for most MBDs [17], vector surveillance is critical and must be strengthened and coordinated on a global scale. Currently, no global surveillance system is in place to track the emergence and spread of MBDs [18, 19]. Increased mosquito surveillance is needed for timely detection of changes in abundance and species diversity, providing valuable knowledge to health authorities and enabling swift mosquito control responses and other public health interventions.

Obtaining field information with traditional mosquito surveillance tools is notoriously costly and time-consuming, and a major drawback of these tools is that they lack scalability. Costs can be significantly reduced by combining citizen science approaches with traditional ones for targeted surveillance [20, 21], and using big data spatial modelling techniques to produce risk maps of vector presence and abundance, human–vector interactions, and disease transmission zones at local or regional scales [22, 23]. Citizen science and the use of digital and networked technologies (Internet, mobile phones) have provided a new dimension for scientific research in the fields of vector ecology, eco-epidemiology, and public health [24, 25].

In the context of MBDs, numerous continuing citizen science surveillance projects (29 projects operating in 16 countries all over the world, including some with wide geographic coverage) [26] have successfully involved members of the public and provided data on mosquito populations. For future improvement, there is a need to continue engaging with stakeholders, researchers, public health agents, industry, and policymakers.

## Context

Mosquito Alert [27] is a citizen science system aimed at investigating and managing disease-carrying mosquitoes. It has been operational since 2014. Initially, most participants were located in Spain, but since 2020 participation has expanded worldwide, particularly in Europe [20, 28]. It uses mobile phones and the Internet to bring together citizens, scientists, and public health authorities to fight against MBDs. Mosquito Alert combines authoritative data with citizen science methodologies for data quality assessment and modeling, enabling large-scale estimates of mosquito population dynamics and the human–mosquito interactions through which MBDs are transmitted across a range of scales.

The dataset presented here was collected through the Mosquito Alert mobile phone application. Citizen scientists provide geolocalized reports and images of targeted mosquito species, breeding sites and biting behavior. Mosquito Alert also includes a module for sending samples to reference research labs in Europe that can be launched when and where considered necessary, allowing these labs to perform vector-specialised identification and screening analyses. In addition, the app collects anonymous information on the geographic distribution of participants to correct for sampling effort biases [20]. The application also includes participant scoring and a notification system that provides scientific and educational content to participants. These features are expected to increase engagement and encourage frequent and extensive participation [29].

The five target species that citizen scientists can report are *Ae. albopictus*, *Ae. aegypti*, *Ae. japonicus*, *Ae. koreicus*, and *Cx. pipiens*. The targeted *Aedes* species are relatively easy to identify in photographs, whereas *Culex pipiens* can be difficult to distinguish from other *Culex* species. App tutorials and communication with citizen scientists are used to aid identification and reporting of the targeted species. Adult mosquito reports containing photos are validated independently by three expert entomologists from the Digital Entomological Network on a private web-based platform, the digital *Entolab*. In addition to these species of interest, expert entomologists also identify other species of mosquitoes (not targeted) and even other insect groups. These identifications are also valuable from an educational perspective, as they help citizen scientists understand the differences between targeted and non-targeted mosquitoes/insects. Note that only the five target species of interest are included in the dataset presented here. Since manual inspection of digital images is not a scalable option, the Mosquito Alert database of expert-validated images has been used to train a deep learning model to find *Ae. albopictus* [30] and the other target species (work in progress). This artificial intelligence system will not only be a helpful pre-selector for the expert validation process but also an automated classifier giving quick feedback to the app participants, which is expected to contribute to long-term motivation.

In this dataset we must differentiate between two periods: the period 2014 to August 2020, and the period September 2020 to 2021. In 2014–2020 the project was mainly focused in Spain, funded by various national sources. Therefore, most of the reports are from Spain. During this period, the system targeted only two invasive species: *Ae. albopictus* and *Ae.*



*aegypti*. The results in Spain during this period and beyond have been valuable. Mosquito Alert has monitored the spread of *Ae. albopictus* in Spain [31, 32], and investigated mosquito species dispersal mechanisms [33]. It was also the source of the first-ever confirmed observation of *Ae. japonicus* in Spain, thus it has served as the basis for estimating that species' distribution in the northern areas of the country [34, 35]. Mosquito Alert also provides direct links between researchers, public health authorities and members of the public, serving as a valuable means for promoting public awareness and education about MBDs.

From September 2020 to 2021, the number of targeted mosquito species was increased to the five mentioned previously, and the project was extended across Europe with the support of European funding (AIM-COST OC-2017-1-22105, CA17108; VEO SC1-BHC-13-2019, 874735). These projects have facilitated the changes needed to increase the number of targeted species, scale the system up to the European level, and promote the development of a Digital Entomological Network of experts, thus boosting the dissemination of activities across Europe to promote data collection and direct interaction with citizen scientists in different countries. In 2020 and 2021, digital citizen science surveillance through Mosquito Alert was carried out in combination with pan-European harmonized field entomological sampling (AIMSurv campaigns [36]) under the framework of the AIM-COST Action. Data outputs of these activities are presented separately in this special collection.

## METHODS

### Study extent

There are no limitations in terms of the geographic areas from which citizens can participate, so data can be sent from all over the world. Nevertheless, Mosquito Alert's main coverage has been in Spain, with increasing coverage in Europe since 2020, mainly in the Netherlands, Italy, and Hungary (Figure 1). The temporal coverage of the dataset is from June 18, 2014 to September 20, 2021 and its temporal distribution is represented in Figure 2. In the dataset presented here, only the five target species are included: *Ae. albopictus*, *Ae. aegypti*, *Ae. japonicus*, *Ae. koreicus*, and *Cx. pipiens*.

### Sampling

There is no predetermined sampling frequency: participants can send as much data as they like, wherever and whenever they choose. Data sampling may be more intense in some periods owing to dissemination events (e.g. project appearances in TV, science fairs, etc.) but is also naturally modulated by mosquito seasonal prevalence and activity patterns.

### Method steps

There are typically five steps to build an occurrence record:

1  An anonymous citizen scientist observes an adult mosquito (dead or alive).
2  Within the Mosquito Alert smartphone application, the citizen scientist answers a short questionnaire with taxonomic and environment-related questions, indicates the location of the observation, attaches one or more photographs (optional), and adds comments (optional).
3  The report is reviewed by members of the Mosquito Alert team to remove any personally identifying information or inappropriate content.



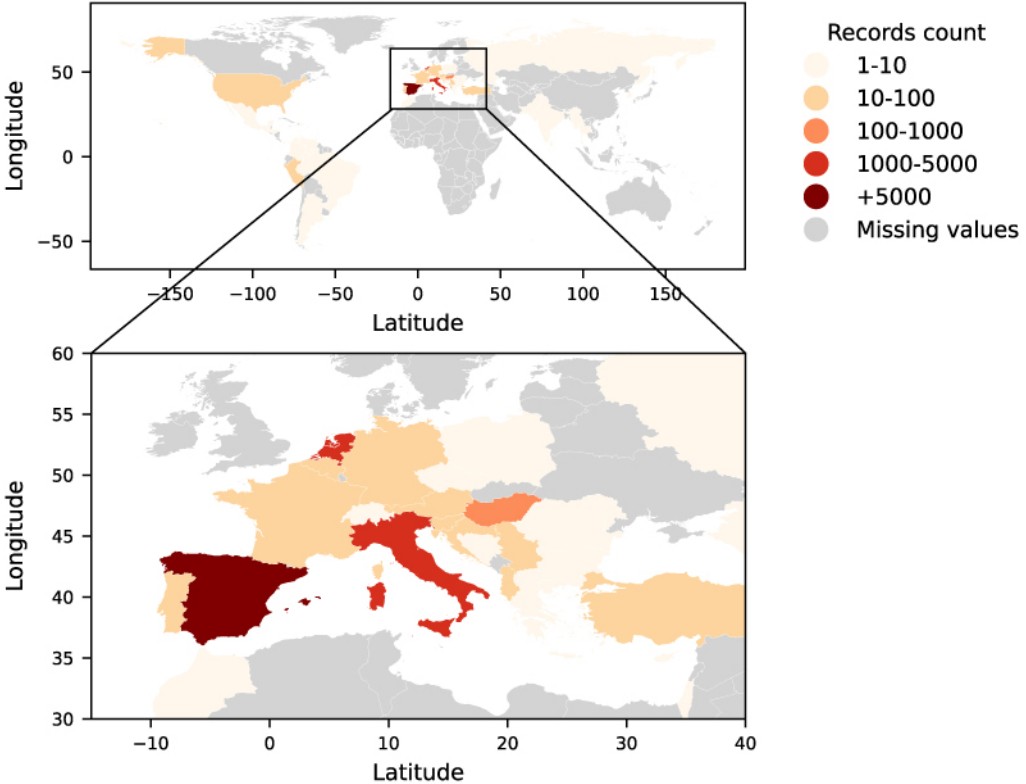

**Figure 1.**  Mosquito Alert occurrence dataset spatial coverage from June 2014 to September 2021.

4   Photographs attached to the report are evaluated independently by three entomologists. Each expert assigns a label to the report, indicating their degree of certainty as to whether the photographs show the target species. A "not sure" label is used if an expert is not able to classify a report. A report is flagged if, for any reason, the report needs further discussion or should be temporarily omitted from the public view. The final taxonomic classification comes from an average of the three expert validations.

5   The report is released into the public domain after validation by the three entomologists, and is reviewed by a senior entomologist who also checks flagged reports. Citizen scientists can include several pictures of the same specimen in asingle report, so one of the three experts is responsible for choosing the final image released to the public domain (public map), which is the one published in the GBIF dataset. The selection criteria is to choose the mosquito image that best represents the observation, or the one most valid for species determination.

## DATA VALIDATION AND QUALITY CONTROL

The Digital Entomology Network comprises several experts, including the so-called 'national supervisor*s*'. National supervisors are national-level coordinators and supervisors within each European country participating in the AIM-COST and VEO projects. In addition, a senior entomologist 'super expert' is in charge of coordinating the whole validation flow and mechanics in the Mosquito Alert system. A manual for the expert validation system is



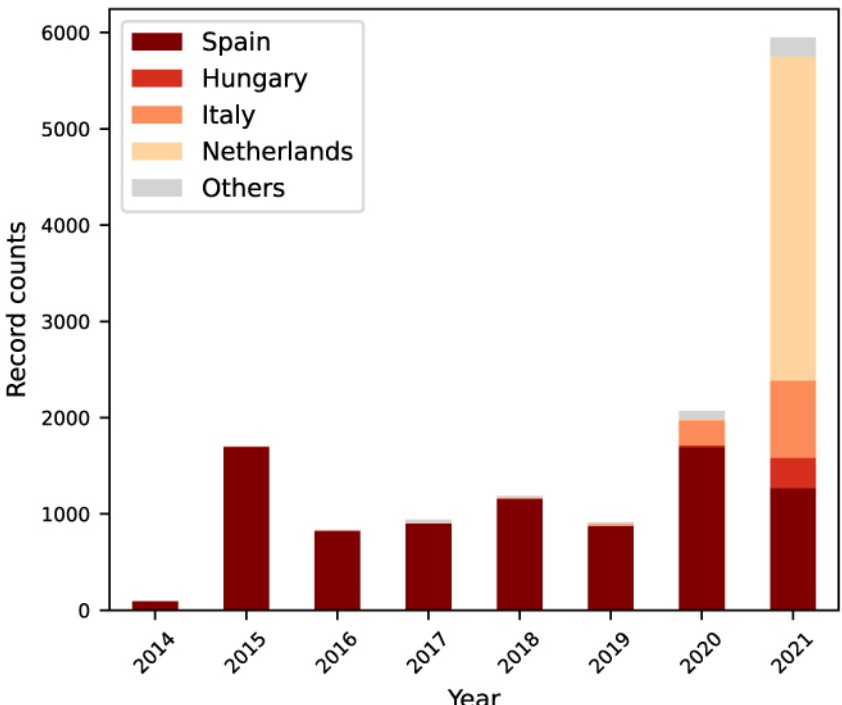

**Figure 2.** Mosquito Alert occurrence dataset temporal coverage.

distributed beforehand to the members of the network and published on the Mosquito Alert website [37], with specifications on the criteria for species determination.

The taxonomic determination of an observation results in two potential outputs indicating the degree of certainty: 'confirmed', when taxonomic features can be clearly seen in the picture(s), and 'probable', when only some characteristic features can be observed. The final taxonomic determination and the relative degree of certainty are computed based on expert validations in two steps:

1 **Selection of most voted category**. The selection for the most voted category is a simple majority selection. For example, assume the following three expert validation assessments: "Probably *Ae. albopictus* | Definitely *Ae. albopictus* | Probably *Ae. aegypti*". The most voted category is *Ae. albopictus,* with two votes. Note that in this step, the 'probably' and 'definitely' qualifications given by each expert are ignored. If there is no majority (i.e., each expert chooses a different taxonomic category) the classification result is considered a 'conflict' and the report is flagged and revised by the super expert.

2 **Certainty value selection**. The certainty labels of the most voted taxonomic category are mapped to integer values such that 1 corresponds to 'probably', and 2 to 'definitely'. The final certainty assessment value is given by averaging the values and rounding them to the nearest integer value with a round half-down strategy. For the above example, the most voted category is *Ae. albopictus* where two values are issued ('probably' and 'definitely') that results in an average value of 1.5. Finally, after rounding the average to 1 the assessment gives a 'probable' *Ae. albopictus* occurrence. If the final result is 2, the certainty degree of the occurrence would be labeled as 'confirmed'. Note that the



rounding half-down strategy implies a conservative approach in the certainty evaluation: if one of the expert expresses doubt, the overall value is decreased.

The validation procedure allows an expert to label a report with 'not sure' in case of pictures with insufficient information. Those records are not included in the current dataset, since only confirmed or probable mosquito records are valid occurrences. For each record, the corresponding entomologist experts who reviewed it are cited by name or by a group label (e.g. institution, team name, etc.). The 'anonymous expert' label is assigned to experts who wish to remain anonymous.

## REUSE POTENTIAL

This dataset and the citizen science system that produced it can reach many entomological (vector) surveillance and management objectives. Firstly, owing to its scalability and large networking capability, Mosquito Alert can be used as an early warning system (EWS) to detect invasive species across scales, from city to continental scales. At local scales, these types of data can help optimize vector control, as citizen scientists provide information about nuisance and presence of mosquitoes in almost real time. Mosquito reduction campaigns may combine top-down strategies of mosquito (larvae) control (undertaken by public health agencies) with bottom-up strategies promoting social action and behavioral change to reduce the proliferation of domestic and peri-domestic breeding sites. Secondly, if combined with other data sources, these data can be used to make risk assessments, such as the characterization of critical areas and seasonal variability for disease risk transmission. They can also be used for data augmentation and calibration in mosquito distribution models of seasonal and inter-annual patterns, as well as and spatial suitability maps. Thirdly, the associated images contribute to efforts to train machine-learning models for image flow optimization procedures in digital-based EWS and mosquito detection and classification.

## DATA AVAILABILITY

The dataset described here is hosted in the *GBIF-Spain* repository [38]. The associated multimedia dataset (mosquito pictures) is available on the *BioImage Archive* repository [39].

## EDITOR'S NOTE

This paper is part of a series of Data Release articles working with GBIF and supported by the Special Program for Research and Training in Tropical Diseases (TDR), hosted at the World Health Organization [40].

## DECLARATIONS
## LIST OF ABBREVIATIONS

AIMs: *Aedes* Invasive Mosquitoes; MBDs: Mosquito-borne diseases; VBDs: Vector-borne diseases; VEO: Versatile emerging infectious disease observatory; WNV: West Nile virus; EWS: Early warning systems.

## ETHICAL APPROVAL

This dataset involves human participation through a mobile phone app from which citizen scientists send text and image data. Participants must accept the Mosquito Alert *User Agreement* [41] in order to use the app, and participation is anonymous.

## CONSENT FOR PUBLICATION

Consent to publish data is stipulated within the Mosquito Alert User Agreement [41], where a consent form is signed by the participant during registration.

## COMPETING INTERESTS

The authors declare that they have no competing interests.

## FUNDING

This work was supported by:

- 2021–2022 Fair Computational Epidemiology (FACE); Plataforma Temática Interdisciplinar PTI+ Salud Global, Consejo Superior de Investigaciones Científicas (CSIC); Grant No.: N/A.
- 2020–2025 Human-Mosquito Interaction Project: Host-vector networks, mobility and the socio-ecological context of mosquito-borne disease; European Research Council (ERC); Grant No.: 853271.
- 2020–2021 Strengthening Barcelona's Defenses Against Disease-Vector Mosquitoes: Automatically Calibrated Citizen-Based Surveillance, Barcelona Ciència; Ajuntament de Barcelona, Institut de Cultura; Grant No.: BCNPC/00041.
- 2020–2024 VEO: Versatile Emerging infectious disease Observatory, H2020 SC1-BHC-13-2019; European Commission (EC); Grant No.: 874735.
- 2020–2025 Preparing for vector-borne virus outbreaks in a changing world: a One Health Approach; Dutch National Research Agenda (NWA); Grant No.: NWA/00686468.
- 2019–2021 Big Mosquito Bytes: Community-Driven Big Data Intelligence to Fight Mosquito-Borne Disease; Fundació "La Caixa", Health Research 2018 "la Caixa" Banking Foundation; Grant No.: HR19-00336.
- 2018–2022 Aedes Invasive Mosquitoes (AIM), COST ACTION OC-2017-1-22105; European Cooperation in Science and Technology (COST); Grant No.: CA17108.
- 2018 Mosquito Alert: programa para investigar y controlar mosquitos vectores de enfermedades como el Dengue, el Chikungunya y el Zika; Fundació "La Caixa"; Grant No.: N/A.
- 2017–2019 Plataforma Integral per al Control de l'Arbovirosis a Catalunya (PICAT); Departament de Salut, Programa PERIS 2016–2020, Generalitat de Catalunya; Grant No.: 00466.
- 2016–2018 Ciència ciutadana per a la millora de la gestió i els models predictius de dispersió i distribució real de mosquit tigre a la Província de Girona; Diputació de Salut de Girona (DIPSALUT); Grant No.: N/A.
- 2016 Nuevas herramientas de participación en ciencia ciudadana: laboratorios de validación y cocreación para AtrapaelTigre.com; Fundación Española para la Ciencia y la Tecnología (FECYT); Grant No.: FCT-15-9515.
- 2016–2017 Mosquito Alert: programa para investigar y controlar mosquitos vectores de enfermedades como el Dengue, el Chikungunya y el Zika; Fundació "La Caixa"; Grant No.: N/A.
- 2016–2017 Ciència ciutadana per a la millora de la gestió i els models predictius de dispersió i distribució real de mosquit tigre a la Província de Girona; Diputació de Salut de Girona (DIPSALUT); Grant No.: N/A.

- 2015–2016 Citizens-based early warning systems for invasive species and disease vectors: The case of the Asian Tiger mosquito; Fundació "La Caixa" and Centre de Recerca Ecològica i Aplicacions Forestals (CREAF); Grant No.: N/A.
- 2014–2016 Invasión del mosquito tigre en España: Salud pública y cambio global; Ministerio de Economía y Competitividad, Plan Estatal I+D+I; Grant No.: CGL2013-43139-R.
- 2014 Diseño e implementación de un sistema ciudadano de alerta y seguimiento del mosquito tigre: ciencia en sociedad (Atrapa el Tigre 2.0); Fundación Española para la Ciencia y la Tecnología (FECYT); Grant No.: FCT-13-701955.

## AUTHOR'S CONTRIBUTIONS

ŽJ-Z: Writing – original draft, Writing – review & editing, Data curation. IS-C: Writing – original draft, Validation. RE: Data curation, Validation. JRBP: Conceptualization, Supervision, Funding acquisition, Software, Data curation, Writing – review & editing. AE: Software, Data curation. JG: Data curation. AO: Conceptualization, Data curation, Project administration. AR-B: Project administration. FS: AIMSurv conceptualization, Resources. ADT: AIMSurv conceptualization, Funding acquisition, Resources. MÁM: AIMSurv conceptualization, Resources. LB: Resources. MK: VEO Conceptualization, Funding acquisition, Resources. FB: Conceptualization, Funding acquisition, Supervision, Writing – review & editing. Mosquito Alert Digital Entomology Network: Validation. Mosquito Alert Community: Investigation.

## ACKNOWLEDGEMENTS

We gratefully acknowledge the work of the entire Mosquito Alert Community, in particular the anonymous citizen scientists who volunteered their time and energy to participate in this project.

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

## DETAILS OF COLLABORATIVE AUTHORS

### • List of authors in Mosquito Alert Digital Entomology Network

Pedro María Alarcón-Elbal,[32] Mikel Alexander González,[15] Maria Angeles Puig,[31] Karin Bakran-Lebl,[5,23] Georgios Balatsos,[27] Carlos Barceló,[16] Mikel Bengoa Paulis,[3] Marina Bisia,[27] Laura Blanco-Sierra,[1] Daniel Bravo-Barriga,[20] Beniamino Caputo,[14] Francisco Collantes,[25] Hugo Costa Osório,[12] Marcela Curman Posavec,[2] Aleksandar Cvetkovikj,[29] Isra Deblauwe,[30] Sarah Delacour,[10] Santi Escartin Peña,[4] Martina Ferraguti,[18] Eleonora Flacio,[19] Hans-Peter Fuehrer,[23] Sandra Gewehr,[9] Filiz Gunay,[35] Rafael Gutiérrez-López,[16] Cintia Horváth,[17] Adolfo Ibanez-Justicia,[8] Perparim Kadriaj,[24] Katja Kalan,[34] Mihaela Kavran,[21] Gábor Kemenesi,[22] Ana Klobucar,[2] Kornélia Kurucz,[22] Eleonora Longo,[14] Sergio Magallanes,[36] Simone Mariani,[31] Angeliki F. Martinou,[6] Rosario Melero-Alcíbar,[37] Antonios Michaelakis,[27] Alice Michelutti,[11] Ognyan Mikov,[28] Tomas Montalvo,[1] Fabrizio Montarsi,[11] Francesca Paoli,[39] Diego Parrondo Montón,[19] Elton Rogozi,[24] Ignacio Ruiz-Arrondo,[7] Francesco Severini,[38] Nikolina Sokolovska,[13] Maria Sophia Unterköfler,[23] Arjan Stroo,[8] Steffanie Teekema,[8] Andrea Valsecchi,[1] Alexander G. C. Vaux,[33] Enkelejda Velo,[24] Carina Zittra[26]

[1] Agencia de Salud Pública de Barcelona (ASPB), Plaça Lesseps 8 entresol, 08023, Barcelona, Spain

[2] Andrija Stampar Teaching Institute of Public Health (ASTIPH), Mirogojska c. 16, 10 000, Zagreb, Croatia

[3] Anticimex Spain (Anticimex), C/ Jesús Serra Santamans, 5, Planta 3, 08174, Sant Cugat del Vallès, Barcelona, Spain

[4] Associació Mediambiental Xatrac (Xatrac), C/ Pius Font i Quer, S/N, 17310, Lloret de Mar, Girona, Spain

[5] Austrian Agency for Health and Food Safety, Division for Public Health (AGES), Währinger Strasse 25a, 1090, Vienna, Austria

[6] British Forces Cyprus, Joint Services Health Unit (JSHU), Cyprus

[7] Center for Rickettsiosis and Arthropod-Borne Diseases, Hospital Universitario San Pedro-CIBIR (CRETAV-CIBIR), C/Piqueras 98, 3° planta, 26006, La Rioja, Spain

[8] Centre for Monitoring of Vectors, National Reference Centre, Netherlands Food and Consumer Product Safety Authority (CMV-NVWA), Geertjesweg 15, 6706 EA, Wageningen, Netherlands

[9] Ecodevelopment S.A. (ECODEV), Thesi Mezaria, PO Box 2420, 57010 Filyro, Greece

[10] University of Zaragoza, Faculty of Veterinary Medicine of Zaragoza, Animal Health Department (UNIZAR), C/ Miguel Servet 177, 50013, Zaragoza, Spain

[11] Istituto Zooprofilattico Sperimentale delle Venezie (IZSVe), Viale dell'Università 10, 35020, Legnaro (Padua), Italy

[12] National Institute of Health, Centre for Vectors and Infectious Diseases Research (INSA-CEVDI), Avenida Padre Cruz, 1649-016, Lisboa, Portugal

[13] PHI Center for Public Health-Skopje (CPH), blv.3rd Macedonian brigade, no.18, Skopje, North Macedonia

[14] Sapienza University, Department Public Health and Infectious Diseases (UNIROMA1), Piazzale Aldo Moro 5, 00198, Rome, Italy

[15] Universidad Iberoamericana (UNIBE), Avenida Francia 129, 10203, Santo Domingo, Dominican Republic

[16] University Balearic Islands, Applied Zoology and Animal Conservation Research Group (UIB), Ctra. Valldemossa km 7.5, 07122, Palma, Spain

[17] University of Agricultural Sciences and Veterinary Medicine of Cluj-Napoca (USAMV-CN), Calea Mănăştur 3-5, Cluj-Napoca, 400372, Romania

[18] University of Amsterdam, Department of Theoretical and Computational Ecology, Institute for Biodiversity and Ecosystem Dynamics (UvA), Science Park 904, 1098XH, Amsterdam, Netherlands

[19] University of Applied Scieces and Arts of Southern Switzerland, Institute of Microbiology (SUPSI), Via Flora Ruchat-Roncati 15, 6850, Mendrisio Switzerland, Switzerland

[20] University of Extremadura, Veterinary Faculty, Department of Animal Health (Uex), Av/ Universidad S/N 10003 Cáceres, Spain

[21] University of Novi Sad, Faculty of Agriculture, Laboratory for Medical and Veterinary Entomology (UNSFA), Trg Dositeja Obradovića 8, 21000, Novi Sad, Serbia

[22] University of Pécs (UP), Ifúság útja 6, 7624, Pécs, Hungary

[23] University of Veterinary Medicine Vienna, Institute of Parasitology (Vetmeduni), Veterinärplatz 1, 1210, Vienna, Austria

[24] Institute of Public Health, Department of Epidemiology and Control of Infectious Diseases, Vectors' Control Unit (IPH), Str: "Aleksander Moisiu", No. 80, Tirana, Albania

[25] Universidad de Murcia, Departamento de Zoología y Antropología Física (UM), Campus de Espinardo, 30100 Murcia, Spain

[26] University of Vienna, Department of Functional and Evolutionary Ecology (UNIVIE), Djerassiplatz 1, 1030, Vienna, Austria

[27] Benaki Phytopathological Institute, Laboratory of Insects and Parasites of Medical Importance (BPI), 8, Stefanou Delta str., 14561 Kifissia, Athens, Greece

[28] National Centre of Infectious and Parasitic Diseases (NCIPD), 26, Yanko Sakazov blvd., 1504, Sofia, Bulgaria

[29] Ss. Cyril and Methodius University in Skopje, Faculty of Veterinary Medicine-Skopje (FVMS), Lazar Pop-Trajkov 5-7, 1000, Skopje, North Macedonia

[30] Institute of Tropical Medicine, Department of Biomedical Sciences, Unit of Entomology (ITM), Nationalestraat 155, 2000, Antwerp, Belgium

[31] Centre d'Estudis Avançats de Blanes (CEAB-CSIC), C/ d'accés a la Cala St. Francesc 14, 17300 Blanes, Girona, Spain

[32] Universidad Cardenal Herrera CEU-CEU Universities, Facultad de Veterinaria, Veterinary Public Health and Food Science and Technology, Department of Animal Production and Health (PASAPTA), C/ Tirant lo Blanc, 7, 46115 Alfara del Patriarca, Valencia, Spain

[33] Medical Entomology, UK Health Security Agency (UKHSA), Porton Down, Salisbury, SP4 0JG, United Kingdom

[34] University of Primorska, Faculty of Mathematics, Natural Sciences and Information Technologies (UP FAMNIT), Glagoljaška ulica 8, 6000, Koper, Slovenia

[35] Hacettepe University, Department of Biology, Ecology Section, Vector Ecology Research Group (HU-VERG), Hacettepe University, Beytepe Campus, 06800, Ankara, Turkey

[36] Estación Biológica de Doñana, Departamento de Ecología de los Humedales (EBD-CSIC), Avda. Américo Vespucio 26, 41092, Sevilla, Spain

[37] Centro de Educación Superior Hygiea (HYGIEA), Av. de Pablo VI, 9, 28223, Pozuelo de Alarcón, Madrid, Spain

[38] Istituto Superiore di Sanità, Department of Infectious Diseases (ISS), Viale Regina Elena, 299, 00161, Roma, Italy

[39] Museo di Scienze di Trento (MUSE), Corso del Lavoro e della Scienza, 3, 38122, Trento, Italy

