## [Reviewer Report]

Upload additional filesDRR-202203-01/form/Review_DRR-202203-01_MosquitoAlert_CJA.pdfReviewer name and names of any other individual's who aided in reviewer Chris ArmitDo you understand and agree to our policy of having open and named reviews, and having your review included with the published papers. (If no, please inform the editor that you cannot review this manuscript.)YesIs the language of sufficient quality?YesPlease add additional comments on language quality to clarify if needed
The data description as provided at the following URL is in English: https://www.gbif.org/dataset/1fef1ead-3d02-495e-8ff1-6aeb01123408. However, the data description as provided at the following URL is in Spanish: https://doi.org/10.15470/t5a1osAre all data available and do they match the descriptions in the paper? NoAdditional Comments1. The GBIF record refers to the following:

“The dataset includes occurrence records of “adult mosquitoes” from 31798 anonymous Mosquito Alert citizen scientists.”

• https://www.gbif.org/dataset/1fef1ead-3d02-495e-8ff1-6aeb01123408

However, the metadata only lists the occurrence records of 13,700 mosquitoes.

2. In addition, there are 40,978 mosquito images in the BioImage Archive dataset and GPS coordinates are provided for these images.

However, the GBIF metadata only lists the occurrence records of 13,700 mosquitoes.

3. In the manuscript, the authors state the following:

• “Mosquito Alert also provided the first record of Ae. (Fredwardsius) vittatus in northwestern Spain and it has contributed tomosquito biodiversity knowledgemore broadly”

However, there is no mention of Aedes vittatus in the GBIF metadata. 
Are the data and metadata consistent with relevant minimum information or reporting standards? See GigaDB checklists for examples <a href="http://gigadb.org/site/guide" target="_blank">http://gigadb.org/site/guide</a>NoAdditional CommentsThere are major discrepancies between the manuscript and the metadata that is available for download from the GBIF dataset.Is the data acquisition clear, complete and methodologically sound?YesAdditional CommentsIs there sufficient detail in the methods and data-processing steps to allow reproduction?YesAdditional CommentsIs there sufficient data validation and statistical analyses of data quality? YesAdditional CommentsIs the validation suitable for this type of data?YesAdditional CommentsIs there sufficient information for others to reuse this dataset or integrate it with other data?YesAdditional CommentsAny Additional Overall Comments to the AuthorRecommendationMajor Revision

---

## [Reviewer Report]

Reviewer name and names of any other individual's who aided in reviewer Rebecca BogerDo you understand and agree to our policy of having open and named reviews, and having your review included with the published papers. (If no, please inform the editor that you cannot review this manuscript.)YesIs the language of sufficient quality?YesPlease add additional comments on language quality to clarify if needed
Are all data available and do they match the descriptions in the paper? YesAdditional CommentsAre the data and metadata consistent with relevant minimum information or reporting standards? See GigaDB checklists for examples <a href="http://gigadb.org/site/guide" target="_blank">http://gigadb.org/site/guide</a>YesAdditional CommentsIs the data acquisition clear, complete and methodologically sound?YesAdditional CommentsIs there sufficient detail in the methods and data-processing steps to allow reproduction?YesAdditional CommentsIs there sufficient data validation and statistical analyses of data quality? YesAdditional CommentsIs the validation suitable for this type of data?YesAdditional CommentsIs there sufficient information for others to reuse this dataset or integrate it with other data?YesAdditional CommentsAny Additional Overall Comments to the AuthorIt is a well written article that explains the dataset. The citizen science dataset has been growing over the past 10 years (or about this), and considerable thought has gone into creating a dataset with quality control features embedded in the design. Starting in Spain, it is growing in contributions and usage around the world. It is an important dataset for vector-borne diseases and will increase in importance as mosquito vectors spread due to human activity and climate change. Very few quality datasets like this one exist. In fact, this may be the only one that has extensive data quality procedures.

Concerning the review criteria listed on page https://gigabytejournal.com/reviewer-information, the article and associated dataset meets all criteria.

I reviewed the site, https://www.ebi.ac.uk/biostudies/studies/S-BIAD249. Images can be downloaded through the BioImage repository.
I don't find this the easiest interface to search for images to view and/or download, but that is not the fault of the Mosquito Alert dataset. The Mosquito Alert dataset does show successful interoperability to make the images available that have gone through a quality control process.

For this site, https://doi.org/10.15470/t5a1os, it looks clear and well organized. I don't read Spanish, so some of the information I didn't understand, but I could see ways to see the data in maps, reports, metadata, and the data. It is well documented and referenced. Considerable thought has gone into the design of the website, and the quality of the data that is open sourced and available for use.

I recommend this article.RecommendationAccept